# Methylene tetrahydrofolate reductase A1298C polymorphisms influence the adult sequelae of chemotherapy in childhood-leukemia survivors

Iris Elens[1,2,3], Sabine Deprez[4,5], Thibo Billiet[5,6], Charlotte Sleurs[5,7], Veerle Labarque[7,8], Anne Uyttebroeck[7,8], Stefaan Van Gool[9], Jurgen Lemiere[2,7], Rudi D'Hooge[1]*

1 Laboratory of Biological Psychology, KU Leuven, Leuven, Belgium, 2 Department of Child and Adolescent Psychiatry, KU Leuven, University Psychiatric Centre Leuven, Leuven, Belgium, 3 Department of Psychiatry, AZ Delta, Roeselare, Belgium, 4 Department of Imaging and Pathology, KU Leuven, Leuven, Belgium, 5 Department of Radiology, University Hospital Leuven, Leuven, Belgium, 6 Icometrix, Leuven, Belgium, 7 Department of Pediatrics, Pediatric Hemato-Oncology, University Hospital Leuven, Leuven, Belgium, 8 Department of Cardiovascular Medicine, KU Leuven, Leuven, Belgium, 9 Immunologisch Onkologisches Zentrum Köln, Köln, Germany

☯ These authors contributed equally to this work.
* rudi.dhooge@kuleuven.be

**Data Availability Statement:** All relevant data are within the manuscript and its Supporting Information files.

## Abstract

This retrospective correlation study investigated the putative link between methylene tetrahydrofolate reductase (MTHFR) A1298C mutations and chemotherapy-related brain function changes in adult childhood-leukemia survivors. To this end, we determined the relationship between the particular MTHFR1298 genotype (AA, AC or CC) of 31 adult childhood-leukemia survivors, and (1) their CSF Tau and phosphorylated Tau (pTau) levels at the time of treatment, (2) their adult performance intelligence quotient (PIQ), and (3) their regional brain connectivity using diffusion magnetic resonance imaging (dMRI) and resting-state functional MRI (rsfMRI). We confirmed that neuropathology markers Tau and pTau significantly increased in CSF of children after intrathecal methotrexate administration. Highest concentrations of these toxicity markers were found during the induction phase of the therapy. Moreover, CSF concentrations of Tau and pTau during treatment were influenced by the children's particular MTHFR1298 genotype. CSF Tau (but not pTau) levels significantly dropped after folinic acid supplementation. At adult age (on average 13.1 years since the end of their treatment), their particular MTHFR1298 genotype (AA, AC or CC) influenced the changes in PIQ and cortical connectivity that we found to be related to their childhood exposure to chemotherapeutics. In summary, we suggest that homozygous MTHFR1298CC individuals are more vulnerable to the adult sequelae of antifolate chemotherapy.

**Funding:** This study was supported by the charity-based Olivia Hendrickx Research Fund (www.olivia.be) in the form of funding awarded to IE, the University of Leuven, and Leuven University Hospital. Icometrix provided support in the form of a salary for TB. The specific roles of this author are articulated in the 'author contributions' section. The funders had no role in the study design, data collection and analysis, decision to publish, or preparation of the manuscript.

**Competing interests:** The authors have read the journal's policy and have the following competing interests: TB is an employee of Icometrix. This does not alter our adherence to PLOS ONE policies on sharing data and materials. There are no patents, products in development or marketed products associated with this research to declare.

# 1. Introduction

Acute lymphoblastic leukemia (ALL) and non-Hodgkin lymphoma (NHL) account for one third of pediatric malignancies. Long-term survival has increased markedly after the introduction of central nervous system (CNS) prophylaxis [1]. Developmentally adverse cranial irradiation was successfully replaced by intrathecal and high-dose intravenous methotrexate, an antifolate agent, but the functional sequelae of this treatment continue to be of concern [2]. Executive functioning, and more specifically, processing speed and cognitive flexibility appear to be typically affected in non-irradiated childhood-leukemia survivors [3–6]. Defects in executive functioning have been linked to defects in white matter integrity, especially in tracts with more prolonged development [7]. Using diffusion tensor imaging (DTI), we previously demonstrated higher fractional anisotropy (FA) and lower orientation dispersion index (ODI) in the left centrum semiovale of childhood-leukemia survivors. Moreover, these structural changes coincided with hypoconnectivity between the inferior temporal gyrus (ITG), and the default mode network (DMN) that is involved in executive functioning [8,9]. Altered functional connectivity of the DMN in non-irradiated childhood-leukemia survivors has been described by others as an adult marker of chemotherapy-induced brain damage [10–12].

However, adult sequelae of chemotherapy remain difficult to predict. Younger age [3,6,13,14]and higher doses of intrathecal methotrexate [15,16] have been linked to more adverse outcomes. Also, it has been demonstrated that CSF levels of the microtubule-associated protein Tau–a marker for acute axonal damage–increase after intrathecal methotrexate in the treated children. This increase is especially prominent during the induction phase, during which patients are not supplied with folinic acid (as opposed to later treatment phases) [17]. In addition, CSF levels of pTau were also elevated, possibly signaling even more pervasive brain impact [18]. We were able to link CSF pTau during treatment to adult PIQ, but failed to observe a correlation between intrathecal methotrexate and CSF pTau 14, even though genetic and/or dietary-induced folate disturbances have been shown to induce Tau hyperphosphorylation [19]. This suggests the involvement of intermediary risk factors that modulate the late effects of methotrexate exposure.

In this report, we therefore investigate A1298C mutations in the gene for methylenetetrahydrofolate reductase (MTHFR), one of the main regulatory enzymes of homocysteine metabolism that catalyzes the conversion of 5,10-methylenetetrahydrofolate to 5-methyltetrahydrofolate [19–21]. A1298C substitution in the MTHFR gene has been linked to increased risk for executive dysfunction in childhood-leukemia survivors, as opposed to the common C677T mutation [22,23]. We therefore hypothesize that the MTHFR1298 genotype of former patients (AA, AC or CC) might influence some of the previously identified outcomes, shown to be vulnerable to methotrexate exposure, such as (1) Tau and pTau CSF levels during treatment [14,16], (2) adult performance intelligence quotient (PIQ), and (3) functional and structural MRI measures, such as FA and ODI in the left centrum semiovale and resting state functional connectivity (RSFC) between the default mode network (DMN) and the inferior temporal gyrus (ITG) [24].

# 2. Methods

## 2.1. Participants and follow-up during treatment

Results were obtained from the same study cohort as reported earlier [14]. In brief, 31 survivors were recruited (ALL: n = 27, NHL: n = 4; mean age at diagnosis 6.4 years; mean age at testing 20.9 years; mean time since end of treatment 13.1 years), who had been treated between 1994 and 2004 according to the EORTC 58881 (1994–1998) or 58951 (1998–2004) protocols.

**Table 1. Demographic data of the participants.**

|  | MTHFR1298CC | MTHFR1298AC | MTHFR1298AA | P** |
|---|---|---|---|---|
| **Demographics** |  |  |  |  |
| n | 5 | 17 | 9 |  |
| Males (%) | 3 (60) | 8 (47) | 3 (33) |  |
| Diagnosis |  |  |  |  |
| T-NHL | 0 | 2 | 2 |  |
| ALL | 5 | 15 | 7 |  |
| Treatment protocol |  |  |  |  |
| EORTC 58881 | 2 | 6 | 4 |  |
| EORTC 58951 | 3 | 11 | 5 |  |
| • VLR | 1 | 1 | 1 |  |
| • AR1 | 4 | 15 | 5 |  |
| • AR2 | 0 | 1 | 3 |  |
| SES* | 36.50 (13–50) | 40.97 (17–55) | 41.81 (13–60.5) | 0.767 |
| Age at diagnosis | 6.8 (1.7–12.6) | 6.6 (2.5–13.6) | 5.8 (2.2–12.7) | 0.364 |
| Time since diagnosis | 15.1 (11.6–17) | 14.9 (11.6–19.4) | 15.6 (13.2–17.3) | 0.636 |
| Total dose IT MTX (mg) | 142 (86–192) | 161 (120–192) | 143 (100–192) | 0.400 |
| Total dose IV MTX (g) | 15 (11–23) | 19.7 (12–35) | 26 (12–67) | 0.315 |

* Socio-economic index score was calculated by multiplying parental occupation scale value by a weight of 5 and parental education scale by 3, and then summing these products as described earlier [8]. High score indicates high socio-economic status.

** Statistical analysis was based on one-way analysis of variance (ANOVA) with group (MTHFR1298AA, AC and CC) as independent factor. Abbreviations: AR: Average risk; IT: Intrathecal; IV: Intravenous; MTHFR: Methylenetetrahydrofolate reductase; MTX: Methotrexate; SES: Socio-economic status, VLR: Very low risk. There were no differences between the genotype groups in terms of socio-economic index and therapeutic.

See Tables 1 and 3 for treatment and demographic data. We could retrace CSF levels in 30 (for Tau) and 26 (for pTau) survivors (see [18], for collection and analysis), but some dossiers were incomplete (see Table 2). The available values were pooled as long as fewer than 2 data points per treatment phase were missing.

The study was approved by the Institutional Ethical Commission of the University Hospital Leuven (UZ Leuven, dossier no. B322201419664). Written informed consent was obtained from all participants (including parents in case of minors). An informed consent form (ICF) for every participant was provided according to the instructions and guidelines of the Commission. These instructions included a template of the form as well as general instructions (e.g., that the form must be drawn up in a language that is clear and understandable to the participant; https://www.uzleuven.be/nl/ethische-commissie-onderzoek/templates-en-interne-richtlijnen-bij-starten-van-dossier-bij-ec-onderzoek/informed-consent-formulier-icf-opstellen). The ICF is understandable to people who are not health professionals and who are unfamiliar with professional or technical language (i.e., understandable for a 12-year-old). Briefly, the ICF informed participants (in their mother tongue) about the aims of the study, content of the examination, and possible risks of participation and discovery of yet unknown neurostructural findings. Participants received further oral clarification about the contents of the informed consent document, and were particularly informed about the possibility to withdraw from the study, at any time, without any need for justification. The instructions of the UZ Leuven Commission are consistent with the recommendations of the European expert group on clinical trials (implementation of EU Regulation No. 536/2014 about clinical trials on medicinal products for human use; https://ec.europa.eu/health/sites/health/files/files/eudralex/vol-10/2017_01_26_summaries_of_ct_results_for_laypersons.pdf

**Table 2. Some of the patient dossiers in this study were incomplete.**

|  | MTHFR1298CC | MTHFR1298AC | MTHFR1298AA |
|---|---|---|---|
| Tau, diagnosis | - | 5 | 1 |
| Tau, LP2 | - | 2 | 1 |
| Tau, LP5 | - | 1 | - |
| Tau, LP6 | - | - | - |
| Tau, LP7 | - | - | - |
| Tau, induction | - | - | - |
| Tau, interval + reinduction | - | - | - |
| Tau, maintenance | - | 2 | - |
| Tau, total treatment | - | 1 | - |
| pTau, diagnosis | 2 | 6 | 4 |
| pTau, LP2 | 1 | 5 | 4 |
| pTau, LP5 | 1 | 2 | 3 |
| pTau, LP6 | 1 | 2 | 3 |
| pTau, LP7 | 1 | 2 | 2 |
| pTau, induction | 1 | 2 | 2 |
| pTau, interval + reinduction | 1 | 2 | 1 |
| pTau, maintenance |  | 3 | - |
| pTau, total treatment | 1 | 2 | 2 |

This table lists the missing Tau and pTau values in the different phases of the therapeutic schedule.

## 2.2. MTHFR genotyping

Obtained from a blood sample in survivors at the time of cognitive testing and image aquisition, DNA segments were amplified by polymerase chain reaction (Promega® PCR kit) using GoTaq® reaction buffer, 1.5 mM $MgCl_2$, 100 ng of genomic DNA, 0.2 μM of forward and reverse primer, respectively and 1.25 U of Taq polymerase, according to the following protocol: (1) 5-min initial denaturation cycle at 95˚C and (2) 35 cycles each of 30 s at 95˚C, 30 s at 58˚C and 30 s at 72˚C. Primers were based on Krull et al. [23]. Resulting DNA fragments of 138bp were sequenced and genotyped at the respective positions using CLC workbench® software.

## 2.3. Cognitive testing and imaging

Adult intelligence was measured using the Wechsler Adult Intelligence Scale (WAIS IV) [25]. MRI scans were acquired at the Leuven University Hospital using a 3T Philips (Achieva) MRI scanner and a 32-channel phased-array head coil. After preprocessing and quality control, the final sample size was 30 for diffusion-weighted magnetic resonance imaging (dMRI) and 28 for resting-state functional MRI (rsfMRI). Image acquisition, selection of imaging outcome measures and image methodology were as previously described [8]. Based on this previous study, higher fractional anisotropy (FA) and lower orientation dispersion index (ODI) in the left centrum semiovale, and lower connectivity between the default mode network and the inferior temporal gyrus (DMN-ITG) were defined as imaging outcome measures.

## 2.4. Statistical analyses and data availability

Graphs were designed in Graphpad Prism 5. Concentrations of biomarkers were logarithmically transformed. Statistical analyses were performed with SPSS software (v20). Paired sample

**Table 3. Treatment schedules slightly differ between protocols EORTC 58881 and 58951.**

| Phase | Drugs | | LP | Duration |
|---|---|---|---|---|
| | **EORTC 58881** | **EORTC 58951** | | |
| Diagnosis | | | 1 | |
| Induction | | | | |
| Induction 1a | Prednisolone | Prednisolone or dexamethasone* | 2–3 | Day 1–35 |
| | Vincristine | Vincristine | | |
| | Daunorubicine | Daunorubicine | | |
| | Single IT | Single (VLR) or triple IT (AR1 and AR2)** | | |
| | E. Coli | E. Coli asparaginase | | |
| | | IV methotrexate + folinic acid (AR2) Cyclophosphamide (AR2)*** | | |
| Consolidation 1b | Single IT | Single (VLR) or triple IT (AR1 and AR2)** | 4–5 | Day 36–62 |
| | Cyclophosphamide | Cyclophosphamide (AR1 and AR2) | | |
| | Cytarabine | Cytarabine | | |
| | 6-Mercaptopurine | 6-Mercaptopurine | | |
| | | E. Coli asparaginase* | | |
| Interval | 6-Mercaptopurine | 6-Mercaptopurine | 6–9 | Day 1–56 |
| | IV methotrexate + folinic acid*** | IV methotrexate + folinic acid*** | | |
| | Single IT | Single (VLR) or triple IT (AR1 and AR2)** | | |
| | Cytarabine* | | | |
| Reinduction | Dexamethasone | Dexamethasone | 10 | Day 1–49 |
| | Vincristine | Vincristine | | |
| | Doxorubicine | Doxorubicine | | |
| | Cyclophosphamide | Cyclophosphamide (AR1 and AR2) | | |
| | Cytarabine | Cytarabine | | |
| | Single IT | Single (VLR) or triple IT (AR1 and AR2)** | | |
| | 6-Thioguanine | 6-Thioguanine | | |
| | E. Coli | E. Coli asparaginase | | |
| Maintenance | 6-Mercaptopurine | 6-Mercaptopurine | 11–15 | 74 weeks |
| | PO Methotrexate | PO Methotrexate | | |
| | Single IT (CNS positive) | Triple IT (AR1 and AR2) | | |
| | IV methotrexate + folinic acid (CNS positive) | IV methotrexate + folinic acid (AR2) E. Coli asparaginase (AR2) | | |
| | | Vincristine* (AR1 and AR2) Prednisolone or dexamethasone* (AR1 and AR2) | | |

* According to randomization

** Single IT: Methotrexate: Less than 1 year: 6 mg, 1–2 year 8 mg, 2–3 years 10 mg, 3 years and more: 12 mg; Triple IT: Methotrexate: Less than 1 year: 6 mg, 1–2 year 8 mg, 2–3 years 10 mg, 3 years and more: 12 mg; Cytarabine: Less than 1 year: 15 mg, 1–2 year 20 mg, 2–3 years 25 mg, 3 years and more: 30 mg; Hydrocortisone: Less than 1 year: 7.5 mg, 1–2 year: 10 mg, 2–3 years: 12.5 mg, 3 years and more: 15 mg.

*** Cyclophosphamide 1000mg/m$^2$, methotrexate 5000mg/m$^2$ + leucovorin 12 mg/m$^2$/6h starts at 36h. Abbreviations: AR: Average risk; CNS: Central nervous system; IT: Intrathecally; IV: Intravenously; PO: Orally; VLR: Very low risk.

t-tests were used to compare CSF biomarker levels between treatment phases and lumbar punctures. One-way ANOVA was used to assess differences in single or pooled CSF biomarker levels with "genotype" as factor (MTHFR1298AA, -AC or -CC), followed by post-hoc least significant difference (LSD) testing. Repeated-measures two-way ANOVA was used to assess the differences in CSF biomarker levels between participant groups with "genotype" as between factor and "time of sample" as repeated measure, followed by post-hoc least significant difference (LSD) testing.

Correlations were calculated as Pearson's r (pooled CSF Tau or pTau, PIQ, FA and ODI at the left centrum semiovale and the DMN-ITG RSFC), whereas correlations including discrete genotypes (AA, AC, CC) were assessed using Spearman's rank correlation coefficients. We used one-sided correlational testing in our sample of former patients that was further subdivided in MTHFR1298 genotype groups. All our hypotheses were indeed distinctly one-sided as it is well established that (1) reduced folate heightens CSF Tau and pTau [17,18,24,25], (2) cognitive performance is reduced in some survivors of childhood leukemia [23,26], (3) MTHFR genotype adversely affects cognitive outcome in child leukemia survivors [22,23], and (4) reduced brain connectivity affects cognitive performance [10].

Raw data that refer to the figures are included as Supplementary Materials. Any further information or data can be obtained from the corresponding author on simple request.

## 3. Results

Highest levels of CSF biomarkers were recorded during the induction phase of the treatment (Fig 1A and 1B). Tau levels were different between "first lumbar puncture (LP1)" and "pooled values of induction phase (LP2-5)", between "LP2-5" and "pooled values of interval + reinduction phase (LP6-10)", and between "LP6-10" and "pooled values of maintenance phase (LP11-15)" (paired sample t-test, t = -4.671, 3.158, 3.046, df = 24, 30, 28, p < 0.001, 0.004, 0.005, respectively). Levels of pTau were different between "LP1" and "LP2-5", between "LP2-5" and "LP6-10", and between "LP6-10" and "LP11-15" (paired sample t-test, t = -5.837, 1.200, 1.069, df = 17, 24, 22, p < 0.001, 0.242, 0.297, respectively). Both CSF Tau and pTau increased after the first intrathecal methotrexate administration, LP1 *versus* LP2 (paired sample t-test, t = -2.985 and -4.128, df = 24 and 17, p = 0.006 and 0.001, respectively). Biomarkers remained elevated during the induction phase, LP1 *versus* LP5 (paired sample t-test, t = -3.519 and -2.524, df = 23 and 16, p = 0.002 and 0.023, respectively for CSF Tau and pTau, respectively). Pooled CSF Tau and p-Tau were strongly correlated (Pearson's r = 0.864, p < 0.001), but only CSF Tau (but not pTau) dropped significantly after the induction phase, P2-5 *versus* P6-10 (paired sample t-test, t = 3.158 and 1.200, df = 30 and 24, p = 0.004 and 0.242, for CSF Tau and pTau, respectively).

Fig 2 also illustrates the parallel baseline shift in biomarker levels between MTHFR1298 genotypes. In fact, baseline pTau values (i.e., prior to intrathecal methotrexate administration) were already different between genotypes, whereas baseline Tau levels were marginally different (Fig 2A and 2B, one-way ANOVA, $F_{2,22}$ = 2.384, p = 0.058 and $F_{2,16}$ = 2.894, p = 0.043 for CSF Tau and pTau, respectively). More specifically, post hoc LSD testing indicated that baseline CSF pTau in MTHFR1298CC genotypes differed significantly from those in AA

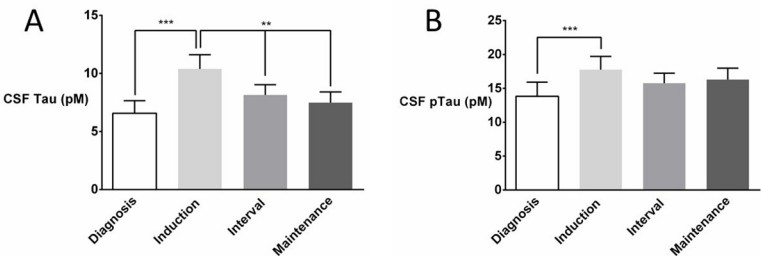

**Fig 1.** After an initial rise during the induction phase, CSF levels of Tau (A), but not those of pTau (B), declined during subsequent treatment phases. Treatment phases (diagnosis, induction, interval and maintenance) are further detailed in Table 2. Paired-sample t-tests were used for post-hoc comparison of CSF biomarker levels between treatment phases with * p < 0.05; ** p<0.01; *** p<0.001. Data are means with SEM (error bars).

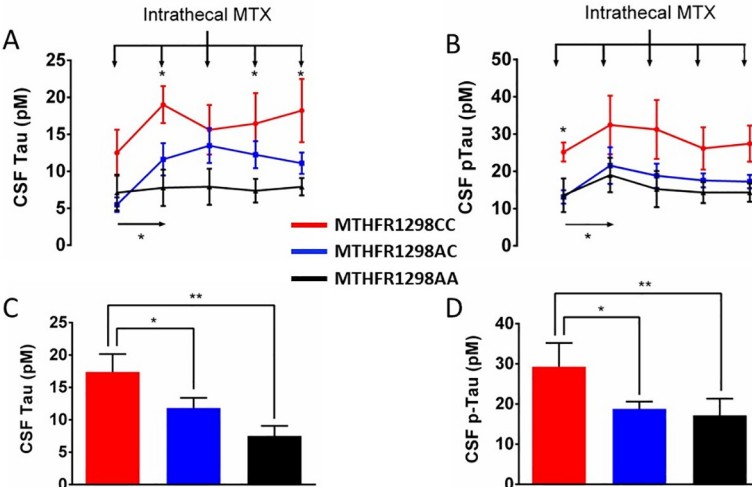

**Fig 2. During the induction phase, CSF levels of Tau and pTau are consistently different between the methylenetetrahydrofolate reductase (MTHFR) genotype groups.** Panels A-B depict mean CSF Tau (A) and pTau (B) levels (± SEM) in subsequent lumbar puncture (LP) samples (LP1: diagnosis, LP2-5: Induction) in the three genotype groups. Vertical arrows at the top of each figure represent the intrathecal methotrexate (MTX) injections after each LP. The first MTX administration produced a significant rise in CSF Tau and pTau levels. Notably, biomarker levels were already higher in the MTHPR1298CC group prior to MTX injection (LP1). Panels C-D depict CSF Tau (C) and pTau (D) levels during the induction phase (LP2-5) in the three genotype groups. Post-MTX, pooled Tau and pTau levels were highest in MTHFR1298CC homozygotes, intermediate in heterozygotes. MTHFR genotype is indicated with color in all panels (red: MTHFR1298CC, blue: MTHFR1298AC, black: MTHFR1298AA). Appropriate ANOVA was followed by post-hoc LSD testing with * p < 0.05; ** p < 0.010; *** p<0.001.

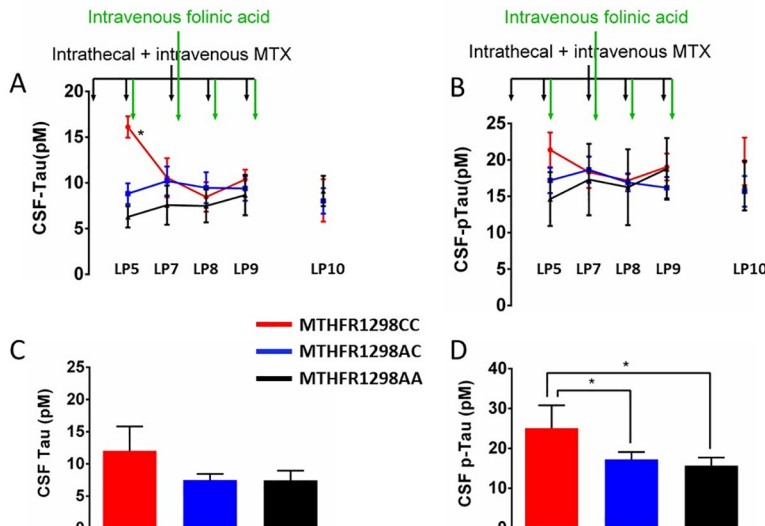

**Fig 3. During the latter treatment phases, intravenous folinic acid administration reduced the highly elevated CSF levels of Tau (but not pTau) in methylenetetrahydrofolate reductase (MTHFR) MTHFR1298CC individuals.** Methotrexate (MTX) treatment was continued (black vertical arrows at the top of each figure). Panels A-B depict CSF Tau (A) and pTau (B) levels during the interval and reinduction phases (lumbar puncture, LP6-10). Initially elevated CSF Tau (but not pTau) levels (LP6), equalized after intravenous folinic acid administration (LP7-10, green vertical arrows). Panels C-D depict CSF biomarker levels during the maintenance phase. CSF pTau differed significantly between genotypes (D), whereas Tau did not (C). MTHFR genotype is indicated with color in all panels (red: MTHFR1298CC, blue: MTHFR1298AC, black: MTHFR1298AA). Appropriate ANOVA was followed by post-hoc LSD with * p < 0.05; ** p < 0.010; *** p< <0.001.

homozygotes (M = 0.273, SD = 0.143, p = 0.036) and in heterozygotes (M = 0.304, SD = 0.128, p = 0.015). However, baseline CSF pTau did not differ between AA homozygotes and heterozygotes (M = 0.031, SD = 0.106, p = 0.386). Fig 2A and 2B furthermore shows that MTHFR1298 genotype significantly influenced biomarker response to the first intrathecal methotrexate injection (two-way ANOVA with genotype and time of sample as factors, LP1 *versus* LP2: $F_{2,22}$ = 3.802 and $F_{2,16}$ = 3.366, p = 0.019 and 0.031 for CSF Tau and pTau, respectively). Post-hoc LSD tests showed significant differences between MTHFR1298CC and both MTHFR1298AA (p = 0.007 and 0.028 for CSF tau and pTau, respectively) and MTHFR1298AC (p = 0.022 and 0.011 for CSF tau and pTau, respectively), but not between the latter two groups (p = 0.19 and 0.38 for CSF Tau and pTau, respectively. There was no time x genotype interaction.

During the induction phase, the MTHFR1298CC group presented the highest concentrations of CSF Tau and pTau (LP2-5), whereas MTHFR1298AC levels were situated in between the two other groups (Fig 2C and 2D, one-way ANOVA, $F_{2,28}$ = 5.795 and $F_{2,23}$ = 3.310 and p = 0.004 and 0.028 for CSF Tau and pTau, respectively), with post-hoc LSD test comparing (1) MTHFR1298AA *versus* MTHFR1298CC genotypes (p = 0.002 and 0.009 for CSF Tau and pTau, respectively), (2) MTHFR1298AC *versus* MTHFR1298CC genotypes (p = 0.057 and 0.037 for CSF Tau and pTau, respectively), and (3) MTHFR1298AC *versus* MTHFR1298AA genotypes (p = 0.012 and 0.117).

Fig 3A and 3B illustrates that CSF Tau, but not pTau, was still elevated at the beginning of the interval phase (LP6, one-way ANOVA, $F_{2,28}$ = 4.306 and $F_{2,22}$ = 1.921, p = 0.012 and 0.085, respectively), post-hoc LSD demonstrated significantly higher CSF Tau in MTHFR1298CC compared to MTHFR1298AA (p = 0.004) and MTHFR1298AC genotypes (p = 0.015), whereas the latter groups did not differ significantly (p = 0.14). The difference in CSF Tau level disappeared after intravenous folinic acid supplementation (one-way ANOVA, $F_{2,27}$ = 1.880, p = 0.086), whereas the significant "genotype x sample" interaction confirms what is evident in the figure, namely that this supplementation only had an effect of the high levels in the MTHFR1298CC group (two-way repeated-measures ANOVA, $F_{2,27}$ = 2.676, p = 0.044). During the maintenance phase CSF Tau no longer differed between the three genotypes, whereas CSF pTau did (Fig 3C and 3D, one-way ANOVA, $F_{2,26}$ = 1.620 and $F_{2,25}$ = 2.970, p = 0.109 and 0.035, respectively). Post-hoc LSD confirmed significantly higher CSF pTau concentrations in MTHFR1298CC compared to MTHFR1298AA (p = 0.012) and MTHFR1298AC genotypes (p = 0.034), but not between MTHFR1298AC and AA genotypes (p = 0.2).

Notably, we found that genotype and CSF levels of Tau as well as pTau were significantly correlated (Spearman's rho = -0.497 and -0.406, p = 0.002 and 0.020; respectively). Moreover, MTHFR1298 genotype was significantly correlated with adult PIQ (one-sided Spearman's rho = 0.340, p = 0.036). Mean PIQ was 93.8, 93.4 and 106.7 in the CC, AC and AA genotype groups, respectively. Fig 4A–4D depicts MRI results that indicate that DMN-ITG RSFC and FA (but not ODI of the left centrum semiovale) correlated significantly with the MTHFR1298 genotype (one-sided Spearman's rho = 0.394, -0.345 and 0.050, p = 0.016, 0.034 and 0.398, respectively).

## 4. Discussion

We examined the effect of MTHFR1298 mutations on methotrexate-induced increases in CSF neurotoxicity markers in children treated for leukemia, but even more importantly, also on their adult PIQ, and MRI measures related to frontal connectivity and executive functioning. We observed a marked increase in CSF Tau and pTau levels in children after the first intrathecal methotrexate administration. Highest concentrations of these markers were measured

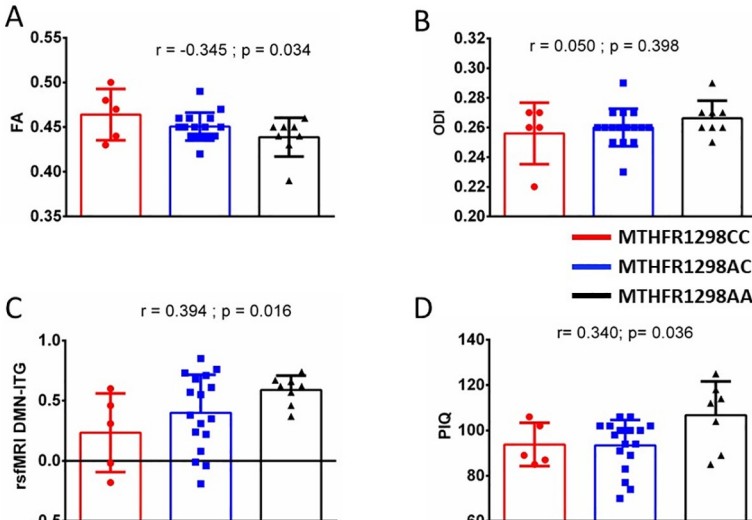

**Fig 4. Methylenetetrahydrofolate reductase (MTHFR) genotype correlates with differences in adult brain connectivity and performal IQ in childhood-cancer survivors.** Panels A-C depict different functional brain connectivity parameters derived from MRI scans, based on our previous study [8], in adult survivors with diverse MTHFR genotype. Fractional anisotropy (FA, panel A), but not orientation dispersion index (ODI, panel B) in the left centrum semiovale, was moderately correlated with MTHFR genotype. Also, functional connectivity between the default mode network (DMN) and the inferior temporal gyrus (ITG) is moderately correlated with MTHFR1298 genotype (C). Finally, performal intelligence quotient (PIQ) is moderately correlated with MTHFR genotype (D). At the top of each figure, r values are 1-sided Spearman's rank correlations (with resp. p values).

during the induction phase, and MTHFR genotype influenced their levels after methotrexate administration and folinic acid supplementation. Intrathecal methotrexate showed the highest impact on CSF toxicity markers in children with MTHFR1298CC genotype compared to MTHFR1298AA, whereas heterozygotes displayed an intermediate response. Moreover, we found indications that the MTHFR genotype still influenced adult PIQ as well as cortical connectivity, many years after treatment.

Methotrexate-induced increases in CSF Tau levels is modulated by the CNS folate status of the children. By inhibiting dihydrofolate reductase, methotrexate depletes the pool of active folates. Besides, methotrexate increases renal folate excretion and inhibits folate uptake into the CSF [27]. The acute rise in CSF Tau after intrathecal methotrexate could be attributed to reduced CSF folate levels during the induction phase [17,18,25,27–30]. Despite continued intrathecal and intravenous methotrexate, CSF Tau significantly declined after systemic folinic acid supplementation, which suggests that the supplementation indeed attenuates the toxicity of the anti-folate regimen. This is in line with previous observations by Winick et al. [31], who reported unexpected signs of acute neurotoxicity, including seizures and other neurological symptoms, in 18% of their children receiving intrathecal MTX. They found that folinic acid supplementation indeed prevented these symptoms in 24 subsequently entered patients.

Notably, folate receptor alfa–one of the two pathways by which folates (including folinic acid) and methotrexate enter the cell–has a higher affinity for folinic acid than for methotrexate [32]. Systemic folinic acid might saturate these receptors, diminish uptake of methotrexate and prevent acute toxicity. We observed that, unlike CSF Tau, CSF pTau failed to respond to folinic acid supplementation. It has been argued that the difference in half-life of CSF Tau and pTau cannot account for this discrepancy [18]. The fact that CSF pTau remained elevated after the induction phase (as opposed to CSF Tau) might be related to the increase in CSF homocysteine during later stages of treatment [28]. This finding is consistent with an earlier report

linking hyperhomocysteinemia with attenuated expression of protein phosphatase 2A, which dephosphorylates pTau to Tau [19].

Our data are also consistent with the general notion that genetic constitution plays a crucial part in chemotherapy-induced brain toxicity [22,23,26,33,34]. MTHFR1298 genotype influenced the levels of CSF Tau and pTau, which could be at least one of the reasons why we previously failed to find a correlation between intrathecal methotrexate dose and CSF pTau levels [14]. In the present study, MTHFR1298CC homozygotes presented with the highest levels of CSF Tau, followed by heterozygotes. Notably, the latter group demonstrated a comparable response to the first intrathecal methotrexate injection as MTHFR1298CC individuals. This suggests that MTHFR1298 polymorphisms may only become relevant when CSF folate levels are challenged. Indeed, the differences between genotypes disappeared after folinic acid supplementation, also indicating that MTHFR1298CC individuals in particular might benefit from this intervention.

Reportedly, polymorphisms in the MTHFR gene neither affect white matter structure or function, nor intelligence scores, in healthy individuals. However, we propose that certain MTHFR genotypes could be more vulnerable than others to the acute and late effects of MTX exposure. There are several indications that the proposed interaction would be physiologically plausible, consistent with some other reports [22,23]. In the present study, CSF pTau (but not Tau) levels continued to differ between MTHFR1298AA and MTHFR1298CC genotypes during the maintenance phase. Additionally, other potentially neurotoxic compounds, such as homocysteine, have been shown to increase after intrathecal MTX [27,28]. Although the proposed interaction might still be debated [34,35], we suggest that the genetically influenced activity of MTHFR could interact with chemotherapy-induced brain folate depletion, which may enhance Tau phosphorylation and concomitant neural toxicity [36,37].

Furthermore, polymorphisms of the MTHFR gene correlated with defects in functional connectivity in the brain of adult childhood-cancer survivors. We report that individuals with the MTHFR1209CC genotype presented the highest FA at the left centrum semiovale. A drug that affects white matter integrity would increase anisotropy especially in brain regions with many crossing fibers [8,38], such as the centrum semiovale, where corpus callosum, corticospinal tract and superior longitudinal fasciculus (SLF) intersect. Such regions have been shown to be especially vulnerable to the acute effects of chemotherapy [7]. Morioka et al. [7] performed DTI in 17 children with ALL and NHL, before and after chemotherapy, and reported defects in corpus callosum and frontal white matter. Such white matter sensitivity has been demonstrated both in vitro [39] and in vivo [38,40]. A possible explanation for this could be that oligodendrocytes are particularly sensitive because of their high metabolic activity. Additionally, suppressed CNS methylation, resulting from folate alterations, may hinder synthesis of myelin basic protein (MBP), a key component of the myelin sheet [21,41,42]. A healthy myelin sheath not only enhances nerve impulse propagation, it also protects axonal fibers against noxious influences [43]. Therefore, defects in the myelination process renders later developing tracts (such as SLF) especially at risk for chemotherapy-induced injury. Since long associative fibers (e.g., SLF) are crucial role for connecting DMN to other brain areas [44], the correlation between DMN-ITG connectivity and MTHFR1298 polymorphisms is of particular interest. Indeed, we observed that genetically determined, decreased MTHFR activity coincided with lower connectivity between DMN and ITG.

## Supporting information

**S1 Data.**
(XLSX)

## Acknowledgments

We are grateful to the participants who contributed to this study, and researchers Elise Bossuyt, Charlotte van Soest, Hannelore Van Gool, Trui Vercruysse, Linde Van den Wyngaert, Femke Pauwels and Lien Solie, who made this work possible. The authors are also grateful to Prof. Dr. Koen Luyckx for statistical advice, Dr. Hugo Vanderstichele and Fujirebio Europe NV for practical and logistic help with collecting and analyzing the CSF samples, Prof. Dr. Jan Cools and Nicole Menten for their assistance in MTHFR genotyping and the pediatric oncology team for their dedicated care for childhood-cancer patients.

## Author Contributions

**Conceptualization:** Iris Elens, Sabine Deprez, Thibo Billiet, Stefaan Van Gool, Jurgen Lemiere, Rudi D'Hooge.

**Data curation:** Stefaan Van Gool.

**Formal analysis:** Iris Elens, Thibo Billiet, Charlotte Sleurs.

**Funding acquisition:** Rudi D'Hooge.

**Methodology:** Iris Elens, Sabine Deprez, Thibo Billiet, Stefaan Van Gool, Jurgen Lemiere.

**Project administration:** Iris Elens.

**Resources:** Stefaan Van Gool, Rudi D'Hooge.

**Software:** Thibo Billiet.

**Supervision:** Sabine Deprez, Veerle Labarque, Anne Uyttebroeck, Jurgen Lemiere, Rudi D'Hooge.

**Validation:** Rudi D'Hooge.

**Writing – original draft:** Iris Elens.

**Writing – review & editing:** Stefaan Van Gool, Jurgen Lemiere, Rudi D'Hooge.

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
