## [Decision Letter · Decision Letter 0]

10 Jun 2020

PONE-D-19-35942

Methylene tetrahydrofolate reductase A1298C polymorphisms influence the adult sequelae of chemotherapy in childhood-leukemia survivors

PLOS ONE

Dear Dr. Iris Elens,

Thank you for submitting your manuscript to PLOS ONE. After careful consideration, we feel that it has merit but does not fully meet PLOS ONE’s publication criteria as it currently stands. Therefore, we invite you to submit a revised version of the manuscript that addresses the points raised during the review process.

We look forward to receiving your revised manuscript.

Kind regards,

Keat Wei Loo

Academic Editor

PLOS ONE

Journal Requirements:

This study was supported by the charity-based Olivia Hendrickx Research Fund (www.olivia.be). The

authors declare no conflicts of interest. We are grateful to the participants who contributed to this

study, and researchers Elise Bossuyt, Charlotte van Soest, Hannelore Van Gool, Trui Vercruysse, Linde

Van den Wyngaert, Femke Pauwels and Lien Solie, who made this work possible. The authors are also

grateful to Prof. Dr. Koen Luyckx for statistical advice, Dr. Hugo Vanderstichele and Fujirebio Europe

NV for practical and logistic help with collecting and analyzing the CSF samples, Prof. Dr. Jan Cools and

Nicole Menten for their assistance in MTHFR genotyping and the pediatric oncology team for their

dedicated care for childhood-cancer patients.

The funders had no role in study design, data collection and analysis, decision to publish, or preparation of the manuscript

4. Please amend your authorship list in your manuscript file to include author Anne Uyttebroeck

The authors have declared that no competing interests exist.

We note that one or more of the authors are employed by a commercial company: Icometrix, Leuven, Belgium.

Additional Editor Comments (if provided):

Author needs to revise as suggested by the reviewer.

Reviewers' comments:

Reviewer's Responses to Questions

**Comments to the Author**

1. Is the manuscript technically sound, and do the data support the conclusions?

Reviewer #1: Partly

2. Has the statistical analysis been performed appropriately and rigorously? 

Reviewer #1: I Don't Know

3. Have the authors made all data underlying the findings in their manuscript fully available?

Reviewer #1: No

4. Is the manuscript presented in an intelligible fashion and written in standard English?

Reviewer #1: Yes

5. Review Comments to the Author

Reviewer #1: Thank-you for the opportunity to review this very interesting manuscript looking at the relationship between folate pathway polymorphisms and cognitive late effects following chemotherapy treatment. The authors are building on studies that proport various alleles in the MTHFR gene place children at increased risk of the side effects of chemotherapy treatment with anti-folate medications, specifically methotrexate. Whilst this is novel research that extends the current literature by including imaging and cognitive outcomes in adults treated as children, I have some concern regarding the strong conclusions made by the authors given the very small number of patients in two of the allele groups.

In the methods the authors provide the number of participants for which Tau measurements were available at each of the treatment timepoints. However, there is no information on what allele group the missing participants were part of. It is therefore possible that the CC or AA groups could be even smaller for some of the analyses undertaken. This is of significant concern particularly for the CC group which only accounts for 5 participants. A breakdown of which group each of the ‘missing’ participants was part of for each of the timepoints is recommended to provide clarity regarding this issue.

The authors also frequently refer to data and descriptions that have been previously published. Although this is a widely accepted practice, it makes the data challenging to accurately place and interpret at times. Providing some minimal information on the sample, imaging approach etc. would be helpful to the reader.

In regards to the results section, I am somewhat confused as to why information that is provided in the results is repeated in the figure legends. I would suggest that the figure legends simply identify what is represented by the figure, rather than repeating the analyses that were undertaken and the outcomes.

We are also not provided with any indication of how these participants perform in regards to their performance IQ and whether any participants exhibit clinically meaningful deficits in this area. This makes the meaningfulness of the relationship described between IQ and the alleles difficult to make sense of.

While I appreciate the approach taken by the authors to investigate this interesting question, the statements made in the conclusion are considered strong given the sample size and range of outcome measures. This is essentially a correlational study, however the wording in the conclusion suggests a causal relationship has been tested. This is somewhat misleading, particularly given that some of the possible relationships may not have been identified for the less common alleles because of a lack of power. Furthermore, some of the relationships identified are weak (e.g. correlation between PIQ and allele), but this has not been explicitly discussed in regards to the implications for the findings. The correspondence between these findings and those of Kamdar etc. who investigated these same alleles is also not covered in any detail in the conclusion, despite the fact that several of the findings are disparate.

There are a number of typos and oversights in the manuscript that are outlined below:

In the hypotheses the authors refer to “read-outs”. It is unclear what this means, and whether they are referring to outcomes.

In Table 1:

• Abbreviations of VLR, AR1 and AR2 are used but not explained. There should be reference to these here even though they are described in the following table.

• SES – the authors make reference to ‘as described earlier’ but there is no description prior to this in the document. If the authors are referring to their previous publication then this should be clearly indicated here, with the reference provided.

In Table 2:

*** are used to refer to methotrexate and cyclophosphamide, but only MTX + leucovorin rescue is highlighted in the footnotes.

Figure 1 – error in line 5, ‘vor’

Figure 2 description line 3 – but slightly not Tau?

Figure 4 – Line 2, missing M in THFR

In discussion – when discussing ODI, the comment is made (but slightly not ODI).

Sentence ‘Particular white matter sensitivity…..’ is not immediately interpretable. Requires further explanation.

Finally, in regards to allowing access to the data, while the authors have responded with 'yes' to this question, it is then stated that the data is only available on request. This requires amendment.

6. PLOS authors have the option to publish the peer review history of their article (what does this mean?). If published, this will include your full peer review and any attached files.

Reviewer #1: No

---

## [Author Response · Author response to Decision Letter 0]

24 Nov 2020

Leuven, September 26, 2020

Dear Editor,

Thank you for your essentially positive response to our manuscript. We addressed all reviewer comments below in our point-by-point reply. We appreciated the opportunity to improve our manuscript.

We hope the revised manuscript is now eligible for publication in PLoS One.

We apologize for the delay in revising the manuscript. As explicated before has the COVID-19 crisis hindered the availability of the first author, currently working as a medical doctor. 

On behalf of all authors,

Iris Elens

Rudi D’Hooge

Biological Psychology, KU Leuven, Belgium

We identified the following major comments:

[…]

 I have some concern regarding the strong conclusions made by the authors given the very small number of patients in two of the allele groups.

Authors’ reply:

Also see comment (3) and (6). We deleted the concluding paragraph and revised the discussion.

 In the methods the authors provide the number of participants for which Tau measurements were available at each of the treatment timepoints. However, there is no information on what allele group the missing participants were part of. It is therefore possible that the CC or AA groups could be even smaller for some of the analyses undertaken. This is of significant concern particularly for the CC group which only accounts for 5 participants. A breakdown of which group each of the ‘missing’ participants was part of for each of the timepoints is recommended to provide clarity regarding this issue.

Authors’ reply:

This information was added to the revised manuscript in table 1. 

 The authors also frequently refer to data and descriptions that have been previously published. Although this is a widely accepted practice, it makes the data challenging to accurately place and interpret at times. Providing some minimal information on the sample, imaging approach etc. would be helpful to the reader.

Authors’ reply:

We are not entirely sure to which descriptions the reviewer refers, we provided more details of several of the cited studies. Also, we thoroughly revised the discussion, toned-down conclusion statements and referred more carefully to published results. As a result, almost every sentence of the discussion was rewritten or revised to address this comment.

 In regards to the results section, I am somewhat confused as to why information that is provided in the results is repeated in the figure legends. I would suggest that the figure legends simply identify what is represented by the figure, rather than repeating the analyses that were undertaken and the outcomes.

Authors’ reply:

On rereading, we found our figure legends indeed quite confusing and often redundant. Legends were completely rewritten and now provide a concise description of the figures.

 We are also not provided with any indication of how these participants perform in regards to their performance IQ and whether any participants exhibit clinically meaningful deficits in this area. This makes the meaningfulness of the relationship described between IQ and the alleles difficult to make sense of.

Authors’ reply:

This information was added to the revised manuscript, in the ‘results’ section. 

 While I appreciate the approach taken by the authors to investigate this interesting question, the statements made in the conclusion are considered strong given the sample size and range of outcome measures. This is essentially a correlational study, however the wording in the conclusion suggests a causal relationship has been tested. This is somewhat misleading, particularly given that some of the possible relationships may not have been identified for the less common alleles because of a lack of power. Furthermore, some of the relationships identified are weak (e.g. correlation between PIQ and allele), but this has not been explicitly discussed in regards to the implications for the findings. The correspondence between these findings and those of Kamdar etc. who investigated these same alleles is also not covered in any detail in the conclusion, despite the fact that several of the findings are disparate.

Authors’ reply:

We agree with the reviewer that our observations should be treated with caution and might even be considered to be preliminary in many respects. We revised the discussion extensively and toned down all our statements that could be perceived as “too bold” for such a correlational study. We also deleted the last sentence of the abstract and revised the penultimate sentence. We do not understand why this reviewer writes that the findings of Kamdar et al. are disparate from ours. In fact, Kamdar et al reported, for instance, that “survivors with MTHFR 1298AC/CC genotypes scored, on average, 13 points lower on TMTB than those with MTHFR 1298AA genotype” (Pediatr Blood Cancer 57: 454–460, 2011).

We identified the following minor comments:

 In the hypotheses the authors refer to “read-outs”. It is unclear what this means, and whether they are referring to outcomes.

 In Table 1:

 Abbreviations of VLR, AR1 and AR2 are used but not explained. There should be reference to these here even though they are described in the following table.

 SES – the authors make reference to ‘as described earlier’ but there is no description prior to this in the document. If the authors are referring to their previous publication then this should be clearly indicated here, with the reference provided.

 In Table 2: *** are used to refer to methotrexate and cyclophosphamide, but only MTX + leucovorin rescue is highlighted in the footnotes.

 Figure 1 – error in line 5, ‘vor’

 Figure 2 description line 3 – but slightly not Tau?

 Figure 4 – Line 2, missing M in THFR

 In discussion – when discussing ODI, the comment is made (but slightly not ODI)

 Sentence ‘Particular white matter sensitivity…..’ is not immediately interpretable. Requires further explanation

 in regards to allowing access to the data, while the authors have responded with 'yes' to this question, it is then stated that the data is only available on request. This requires amendment.

 Cecilia Papp wrote for PLOS ONE: "From all participants (including parents in case of minors), written informed consent was obtained." could you please also add the consent statement to your ethics statement in the methods section of your manuscript.

This information had been added to the manuscript. 

All these comments have been addressed.

---

## [Editor Report · Decision Letter 1]

4 Jan 2021

PONE-D-19-35942R1

Methylene tetrahydrofolate reductase A1298C polymorphisms influence the adult sequelae of chemotherapy in childhood-leukemia survivors

PLOS ONE

Dear Dr. Rudi D’Hooge,

Thank you for submitting your manuscript to PLOS ONE. After careful consideration, we feel that it has merit but does not fully meet PLOS ONE’s publication criteria as it currently stands. Therefore, we invite you to submit a revised version of the manuscript that addresses the points raised during the review process.

ACADEMIC EDITOR: Please refer to the Additional Editor Comments listed below.

We look forward to receiving your revised manuscript.

Kind regards,

Keat Wei Loo

Academic Editor

PLOS ONE

Additional Editor Comments (if provided):

Some minor errors detected" their MTHFR1298 genotype influenced changes in PIQ". It is unclear that which MTHFR genotype are they referring to. It is unclear that which MTHFR genotype are they referring to. It is also unclear what is missing data Tau 2, 7, ....listed in the table. It is also unclear what is missing data Tau 2, 7, ....as listed in the table. This sentence is unclear "They were made aware that they could withdraw at any time from the study, without justification as to why they did so. ". Typo "PIQ was 93.800, 93.411 and 106.714 for CC, AC and AA genotype, respectively' & 'We report that individuals with the MTHFR1209CC genotype presented the highest FA at the left centrum semiovale' & 'This is in line with previous observations. Winick et al.33 reported ........."

---

## [Author Response · Author response to Decision Letter 1]

18 Mar 2021

Rebuttal letter – revision R3

Manuscript PONE-D-19-35942R1: “Methylene tetrahydrofolate reductase A1298C polymorphisms influence the adult sequelae of chemotherapy in childhood-leukemia survivors” by Elens et al.

Rebuttal letter by Iris Elens and Rudi D’Hooge, on behalf of all co-authors

University of Leuven, Belgium

We revised our manuscript according to the following “Additional Editor Comments”:

Some minor errors detected" their MTHFR1298 genotype influenced changes in PIQ". It is unclear that which MTHFR genotype are they referring to. It is unclear that which MTHFR genotype are they referring to. It is also unclear what is missing data Tau 2, 7, ....listed in the table. It is also unclear what is missing data Tau 2, 7, ....as listed in the table. This sentence is unclear "They were made aware that they could withdraw at any time from the study, without justification as to why they did so. ". Typo "PIQ was 93.800, 93.411 and 106.714 for CC, AC and AA genotype, respectively' & 'We report that individuals with the MTHFR1209CC genotype presented the highest FA at the left centrum semiovale' & 'This is in line with previous observations. Winick et al.33 reported ........."

Not all of these comments were clear to us, but we accept that some of these parts were not written as clearly as possible.

Changes in the revised manuscript R3:

(1) It is also unclear what is missing data Tau 2, 7, ....listed in the table. It is also unclear what is missing data Tau 2, 7, ....as listed in the table.

We accept that this was not very clear. We therefore removed these numbers of missing data from Table 1, and included a new Table 2 that lists the different missing data from the incomplete patient dossiers. We also rewrote the corresponding text in the methods section as well.

(2) This sentence is unclear "They were made aware that they could withdraw at any time from the study, without justification as to why they did so.”

We were not certain what was meant here, but we rewrote this sentence and included additional information about the ICF and the IC procedures at the UZ Leuven university hospital.

(3) Typo "PIQ was 93.800, 93.411 and 106.714 for CC, AC and AA genotype, respectively' & 'We report that individuals with the MTHFR1209CC genotype presented the highest FA at the left centrum semiovale' & 'This is in line with previous observations. Winick et al.33 reported ........."

We rewrote these sentences.

---

## [Editor Report · Decision Letter 2]

5 Apr 2021

Methylene tetrahydrofolate reductase A1298C polymorphisms influence the adult sequelae of chemotherapy in childhood-leukemia survivors

PONE-D-19-35942R2

Dear Dr. D'Hooge,

We’re pleased to inform you that your manuscript has been judged scientifically suitable for publication and will be formally accepted for publication once it meets all outstanding technical requirements.

Kind regards,

Roland A Ammann, M.D.

Academic Editor

PLOS ONE
---

## [Editor Report · Acceptance letter]

22 Apr 2021

PONE-D-19-35942R2 

Methylene tetrahydrofolate reductase A1298C polymorphisms influence the adult sequelae of chemotherapy in childhood-leukemia survivors 

Dear Dr. D'Hooge:

I'm pleased to inform you that your manuscript has been deemed suitable for publication in PLOS ONE. Congratulations! Your manuscript is now with our production department. 

Kind regards, 

on behalf of

Professor Roland A Ammann 

Academic Editor

PLOS ONE